# Self-Routing Capsule Networks

**Taeyoung Hahn**     **Myeongjang Pyeon**     **Gunhee Kim**
Seoul National University, Seoul, Korea
{taeyounghahn,mjpyeon,gunhee}@snu.ac.kr
http://vision.snu.ac.kr/projects/self-routing

## Abstract

Capsule networks have recently gained a great deal of interest as a new architecture of neural networks that can be more robust to input perturbations than similar-sized CNNs. Capsule networks have two major distinctions from the conventional CNNs: (i) each layer consists of a set of capsules that specialize in disjoint regions of the feature space and (ii) the routing-by-agreement coordinates connections between adjacent capsule layers. Although the routing-by-agreement is capable of filtering out noisy predictions of capsules by dynamically adjusting their influences, its unsupervised clustering nature causes two weaknesses: (i) high computational complexity and (ii) cluster assumption that may not hold in the presence of heavy input noise. In this work, we propose a novel and surprisingly simple routing strategy called *self-routing*, where each capsule is routed independently by its subordinate routing network. Therefore, the agreement between capsules is not required anymore, but both poses and activations of upper-level capsules are obtained in a way similar to Mixture-of-Experts. Our experiments on CIFAR-10, SVHN, and SmallNORB show that the self-routing performs more robustly against white-box adversarial attacks and affine transformations, requiring less computation.

## 1   Introduction

In the past years, deep convolutional neural networks (CNNs) have become the de-facto standard architecture in image classification tasks, thanks to their high representational power. However, an important yet unanswered question is whether deep networks can truly generalize. Well-trained networks can be catastrophically fooled by the images with carefully designed perturbations that are even unrecognizable by human eyes [23, 29, 37, 38]. Furthermore, natural, non-adversarial pose changes of familiar objects are enough to trick deep networks [1, 8]. The later is more depressing since natural pose changes are universal in the real world.

Some research [5, 15] has argued that neural networks should aim for *equivariance*, not *invariance*. The reasoning is that, by preserving variations of an entity in a group of neurons (equivariance) rather than only detecting its existence (invariance), it would be easier to learn the underlying spatial relations and thus yield better generalization. Following this argument, a new network architecture called *capsule networks* (CapsNets) and a mechanism called *routing-by-agreement* have been introduced [16, 34]. In this design of networks, each capsule contains a *pose* (or *instantiation parameters*) for encoding patterns of its responsible entity. Active capsules in one layer make pose predictions for capsules in the next layer via transformation matrices. Then, the routing algorithm finds a center-of-mass among the predictions via iterative clustering and ensures that only the majority opinion is passed down to the next layer.

While the routing-by-agreement [16, 34] has shown to be effective, its unsupervised clustering nature causes two inherent weaknesses. First, it requires repeatedly computing means and membership scores of prediction vectors. This makes CapsNets much computationally heavier than one-pass

feed-forward CNNs. Second, it makes assumptions on cluster distributions of predictions. This might not be a problem if the training of CapsNets successfully learns to fit the weights to the assumptions. However, it is likely that the routing-by-agreement tends to fail when a number of prediction vectors become noisy so that they are clustered in an unexpected form.

In this work, we aim to overcome the above limitations by proposing a new and surprisingly simple routing strategy that does not involve agreement anymore. In our algorithm, the contribution of a lower-level capsule to a higher-level capsule is determined by its activation and the routing decision by its subordinate routing network. We refer to this design of routing as *self-routing*. To the best of our knowledge, there is no previous literature on removing the routing-by-agreement from CapsNets.

Our method is motivated by the structural resemblance between CapsNets and Mixture-of-Experts (MoE) [18, 7, 36, 20]. They are similar in that their composing units (*i.e.* capsules and experts) specialize in different regions of input space and that their contributions are adjusted differently per example. One key difference is that, in CapsNets, gating values are dynamically adjusted to suppress potentially unreliable submodules via the routing-by-agreement. However, if the robustness of CapsNets can be retained without the routing-by-agreement, then we might be able to safely remove the unsupervised clustering part that causes the two aforementioned weaknesses.

For evaluation, we compare our self-routing to the two most prominent routing-by-agreement, dynamic [34] and EM [16] routing on CIFAR-10 [22], SVHN [42] and SmallNORB [24] datasets. We compare not only classification accuracies but also robustness under adversarial attacks and viewpoint changes. For fairness, we use the same CNN base architectures (*e.g.* 7-layer CNN and ResNet-20) and replace only the last layers of the original networks to respective capsule layers.

Compared to the previous routing-by-agreement methods [16, 34], the self-routing achieves better classification performance in most cases while using significantly fewer computations in FLOPs. Moreover, it shows stronger robustness under both perturbations of adversarial attacks and viewpoint changes. We also show that our self-routing benefits more the CapsNets from the increase in model sizes (*i.e.* wider capsule layers), while the previous methods often degrade.

## 2   Related Work

**Capsule networks**. Recently, capsule networks have been actively applied to many domains, such as generative models [19], object localization [27], and graph networks [40], to name a few. Hinton *et al*. [15] first introduced the idea of capsules and equivariance in neural networks. In their work, autoencoders are trained to generate images with the desired transformation; yet the model requires transformation parameters to be supplied externally. Later, Sabour *et al*. [34] proposed a more complete model in which transformations are directly learnable from the data. To control the information flow between adjacent capsule layers, they employed a mechanism named *dynamic routing*. Since then, alternative routing methods have been suggested. Hinton *et al*. [16] proposed to use Gaussian-mixture clustering. Bahadori *et al*. [3] made convergence faster via eigendecomposition of prediction vectors. Wang *et al*. [41] formalized the routing process to suggest a theoretically refined version. Li *et al*. [25] approximated the routing process with the interaction between master and aide branches. Compared to all of the previous work, our routing approach is free from *agreement* but focuses on its ability of mixture-of-experts.

**Mixture-of-experts.** There have been many attempts to incorporate mixture-of-experts (MoE) into deep network models. Eigen *et al*. [7] stacked multiple layers of MoE to create an exponentially increasing number of inference paths. Shazeer *et al*. [36] used sparsely-gated MoE between stacked LSTM layers to expand model capacity with only a minor loss in computational efficiency. In [30], architectural diversity is added by allowing experts to be an identity function or a pooling operation. Kirsch *et al*. [21] modularized a network so that neural modules can be selected on a per-example basis. In [33], a routing network is trained to choose appropriate function blocks for the input and task. This work interprets the origin of CapsNet's strengths as the behavior of MoE, and such perspective leads to our self-routing design.

**CNN fragility.** Despite the great success of CNNs, many recent studies have raised concerns about their robustness [8, 9, 14]. Unlike humans, CNNs easily yield incorrect answers when presented with rotated images [8, 9]. Surprisingly, little improvement is observed in terms of noise robustness even for recent deep CNNs that are highly successful on image classification tasks [14]. However,

their fragility may be hard to overcome by data augmentation techniques [9]. There are also a number of methods called *adversarial attacks*, that fool CNNs by creating images whose fabrication is hardly perceivable even to human [23, 29, 38, 37]. In this work, we evaluate the robustness of CapsNets, which are proposed as an alternative or a supplement for deep CNNs. In section 5, we demonstrate that augmenting only one routing layer structured by capsules can significantly improve the robustness against adversarial attacks and affine transformations.

## 3 Preliminaries

We first review the basics of capsule networks and the two most popular routing algorithms: dynamic [34] and EM routing [16].

### 3.1 Capsule Formulation

A capsule network [16, 34] is composed of layers of capsules. Let $\Omega_l$ denote the sets of capsules in layer $l$. Each capsule $i \in \Omega_l$ has a pose vector $\mathbf{u}_i$ and an activation scalar $a_i$. In addition, a weight matrix $\mathbf{W}_{ij}^{\text{pose}}$ for every capsule $j \in \Omega_{l+1}$ predicts pose changes: $\hat{\mathbf{u}}_{j|i} = \mathbf{W}_{ij}^{\text{pose}} \mathbf{u}_i$. The pose vector of capsule $j$ is a linear combination (or together with an activation function) of the prediction vectors: $\mathbf{u}_j = \sum_i c_{ij} \hat{\mathbf{u}}_{j|i}$, where $c_{ij}$ is a routing coefficient determined by an routing algorithm. In the convolutional case, capsules within $K \times K$ neighborhood in $\Omega_l$ define a capsule in $\Omega_{l+1}$ where $K$ is the kernel size. The formulation of capsules varies according to the design of the routing algorithm. In dynamic routing [34], the pose is defined as a vector, and its length is used as its activation. In EM routing [16], the pose is defined as a matrix, and the activation scalar is separately defined. In our method, we use a vector for the pose with separated activation scalar.

### 3.2 Dynamic and EM Routing

A capsule is activated when multiple predictions by lower-level capsules agree. In other words, the activation depends on how tight the prediction vectors are clustered. The routing coefficients from a lower-level capsule to all upper-level capsules sum to 1 (*e.g.* $\sum_j c_{ij} = 1$), and are iteratively adjusted so that the lower-level capsule $i$ has more influence on the upper-level capsule $j$ when $i$-th prediction is close to the mean of the predictions that $j$ receives. We below review how the two most popular routing methods compute the routing coefficient $c_{ij}$ from capsule $i$ to $j$.

In dynamic routing [34], the cosine similarity is used to measure the agreement. The routing logits $b_{ij}$ are initialized to 0 and adjusted iteratively by the following equation:

$$b_{ij}^{(t+1)} \leftarrow b_{ij}^{(t)} + \hat{\mathbf{u}}_{j|i} \cdot \mathbf{u}_j^{(t)} \quad \text{for } t = 1, \cdots k, \tag{1}$$

where $t$ is the iteration number. The routing coefficients $c_{ij}$ are obtained by applying softmax to $b_{ij}$ along $j$-th dimension. $\mathbf{u}_j^{(t)}$ is obtained by applying a non-linear squash function: $f_{squash}(\mathbf{s}) = \frac{||\mathbf{s}||^2}{1+||\mathbf{s}||^2} \frac{\mathbf{s}}{||\mathbf{s}||}$ to $\sum_i c_{ij}^{(t)} \hat{\mathbf{u}}_{j|i}$. Then, $c$ and $\mathbf{u}_j$ are updated alternatively for $k$ iterations.

In EM routing [16], it is assumed that the probability density of $\hat{\mathbf{u}}_{j|i}$ follows the $j$'s Gaussian model with mean $\boldsymbol{\mu}_j$ and variance $\text{diag}[\sigma_{j,1}^2, \sigma_{j,2}^2, \cdots, \sigma_{j,h}^2]$ where $h$ is the dimension of $\mathbf{u}_j$ and $\hat{\mathbf{u}}_{j|i}$: $\hat{\mathbf{u}}_{j|i} \sim \mathcal{N}(\boldsymbol{\mu}_j, \text{diag}[\sigma_{j,1}^2, \sigma_{j,2}^2, \cdots, \sigma_{j,h}^2])$ for all $i \in \Omega_l$. The following iterative routing process is conducted with $a_j^{(1)}$ initialized to 0 and for each iteration $t = 1, \cdots, k$,

$$a_j^{(t)}, \boldsymbol{\mu}_j^{(t)}, \boldsymbol{\sigma}_j^{(t)} \leftarrow \texttt{M} - \texttt{step}(a_i, c_{ij}^{(t)}, \hat{\mathbf{u}}_{j|i}), \quad c_{ij}^{(t+1)} \leftarrow \texttt{E} - \texttt{step}(a_j^{(t)}, \boldsymbol{\mu}_j^{(t)}, \boldsymbol{\sigma}_j^{(t)}), \tag{2}$$

where $\text{cost}_j = \sum_{i \in \Omega_l} c_{ij}(\beta_u - \log p(\hat{\mathbf{u}}_{j|i}))$, $a_j^{(t)} = \text{sigmoid}(\lambda(\beta_a - \text{cost}_j^{(t)}))$, $a_j = a_j^{(k)}$ and $\mathbf{u}_j = \boldsymbol{\mu}_j^{(k)}$. $\beta_a$ and $\beta_u$ are learned discriminatively and $\lambda$ is a hyperparameter that increases during training with a fixed schedule.

## 4 Approach

In section 4.1, we discuss the two distinctive strengths of CapsNets. In section 4.2, we propose our approach of *self-routing*, whose motivation is to maximize the merits of CapsNets while minimizing undesired side effects.

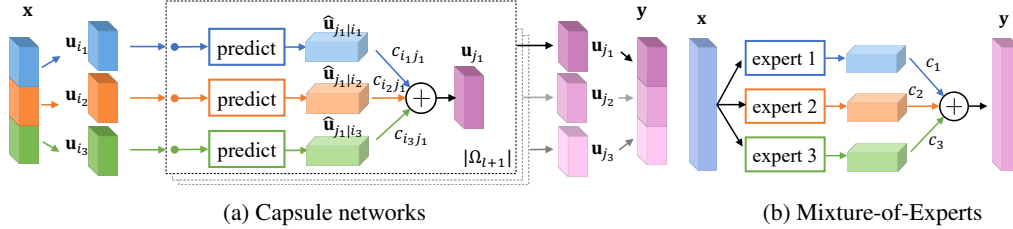

(a) Capsule networks           (b) Mixture-of-Experts

Figure 1: Comparison between (a) capsule networks and (b) mixture-of-experts. Given routing coefficients, the only computational difference is that capsule networks use disjoint input representations for each capsule unit.

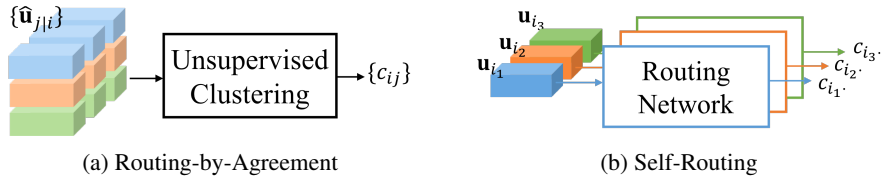

(a) Routing-by-Agreement          (b) Self-Routing

Figure 2: Conceptual comparison between (a) routing-by-agreement and (b) our proposed self-routing. In self-routing, subordinate routing networks $W_i^{\text{route}}$ fed pose vectors $\mathbf{u}_i$ are used to obtain routing coefficients $c_{ij}$ rather than unsupervised clustering on prediction vectors $\hat{\mathbf{u}}_{j|i}$.

## 4.1 Motivation

We view that CapsNets have two distinctive characteristics compared to the conventional CNNs: (i) the behavior of mixture-of-experts (MoE) and (ii) the noise filtering mechanism.

**Behavior of MoE.** Capsules specialize in disjoint regions of feature space and make multiple predictions based on information available to them for the regions. In each capsule layer, this structure naturally forms an ensemble of submodules that are activated differently per example, in a way similar to Mixture-of-Experts (MoE) [18, 7, 36, 20]. Compared to MoE, the division of labor is more explicit as different capsules do not share the same feature space (see Figure 1). Nonetheless, the discriminative power of each capsule may not be as strong as in CNNs, where the entire feature map in each layer is utilized to produce a single output of the layer. However, by aggregating predictions from weaker modules that have different parameters, it effectively prevents overfitting and thus can reduce the output variance with respect to small input variations.

**Noise filtering mechanism.** In CapsNets, initial gating values (*i.e.* activation scalars) of capsules are dynamically adjusted. The routing-by-agreement ensures that the predictions far from general consensus have a lesser influence on the output of each layer. In other words, the process can effectively filter out the contribution of submodules having possibly noisy information. Additionally, output capsules of which predictions have high variance are further suppressed. On the other hand, CNNs have no mechanism of segmenting potentially noisy channels from unnoisy ones.

In this work, we aim to design a routing method that mainly focuses on the first characteristic of CapsNets. The second property is beneficial but brought by the agreement-based routing, which unfortunately causes two critical side effects: (i) high computational complexity and (ii) assumptions on cluster distribution of prediction vectors. Specifically, the previous routing methods assume a spherical or normal distribution of prediction vectors, which is unlikely to hold due to high variability and noisiness of real-world data. In fact, clustering noisy data is still a challenging task. Hence, the key to our intuition is to remove the notion of agreement from the routing process but introduce a learnable routing network for each capsule instead (section 4.2). Although the clustering in the agreement-based routing can help to reduce the variance of output, we empirically observe that, even without the routing-by-agreement, the simple weight average of the new routing method can have a similar effect. That is, the unreliability of a single prediction can be mitigated by *ensemble averaging* since the errors of the submodules (capsules) average out to provide a stable combined output.

## 4.2 Self-Routing

We name the CapsNet model with our proposed self-routing as *SR-CapsNet*. Figure 2(a)–(b) illustrate the high-level difference between the self-routing and the routing-by-agreement. In the self-routing, each capsule determines its routing coefficients by itself without coordinating the agreement with peer capsules. Instead, each capsule is endowed higher modeling power by a subordinate *routing network*. Following the MoE literature [36], we design the routing network as single-layer perceptrons, although it is straightforward to extend it to an MLP. The routing coefficients also work as the predicted activations of output capsules. That is, an upper-level capsule is more likely to be activated if more capsules have high routing coefficients to it.

The self-routing involves two learnable weight matrices, $\mathbf{W}^{\text{route}}$ and $\mathbf{W}^{\text{pose}}$, which are used to compute routing coefficients $c_{ij}$ and predictions $\hat{\mathbf{u}}_{j|i}$, respectively. Each layer of the routing network multiplies the pose vector $\mathbf{u}_i$ by a trainable weight matrix $\mathbf{W}_i^{\text{route}}$ and outputs the routing coefficients $c_{i*}$ via a softmax layer. The routing coefficients $c_{ij}$ are then multiplied by the capsule's activation scalar $a_i$ to generate weighted votes. The activation $a_j$ of an upper-layer capsule is simply the summation of the weighted votes of lower-level capsules over spatial dimensions $H \times W$ (or $K \times K$ when using convolution). In summary,

$$c_{ij} = \text{softmax}(\mathbf{W}_i^{\text{route}}\mathbf{u}_i)_j, \quad a_j = \frac{\sum_{i \in \Omega_l} c_{ij}a_i}{\sum_{i \in \Omega_l} a_i}. \tag{3}$$

The pose of the upper-layer capsule is determined by the weighted average of prediction vectors:

$$\hat{\mathbf{u}}_{j|i} = \mathbf{W}_{ij}^{\text{pose}}\mathbf{u}_i, \quad \mathbf{u}_j = \frac{\sum_{i \in \Omega_l} c_{ij}a_i\hat{\mathbf{u}}_{j|i}}{\sum_{i \in \Omega_l} c_{ij}a_i}. \tag{4}$$

## 5 Experiments

In experiments, we focus on comparing our self-routing scheme with the two most important agreement-based routing algorithms from multiple perspectives. We first evaluate the image classification performance (section 5.2). We then compare the robustness against unseen input perturbations, since such generalization abilities have been the key motivation of CapsNets. Especially, we experiment the robustness against adversarial examples (section 5.3) and viewpoint changes by affine transformation (section 5.4). Our full source code is available at http://vision.snu.ac.kr/projects/self-routing.

### 5.1 Experimental Settings

**Datasets**. Following CapsNet literature, we mostly use two classification benchmarks of CIFAR-10 [22] and SVHN [42] and additionally SmallNORB [24] for the affine transformation tests. During training on CIFAR-10 and SVHN, we augment using random crop, random horizontal flip, and normalization. For SmallNORB, we follow the setting of [16]; we downsample training images to $48 \times 48$, randomly crop $32 \times 32$ patches, and add random brightness and contrast. Test images are center cropped after downsampled.

**Architectures**. For a fair comparison, we let all routing algorithms share the same base CNN architecture. We choose ResNet-20 [13] designed for CIFAR-10 classification for the following two reasons. First, the ResNet is one of the best performing CNN architectures in various computer vision applications [4, 11, 26, 31, 32]. It would be interesting to verify whether employing capsule structures can benefit mainstream CNNs. Second, the ResNet is mostly composed of Conv layers, which makes it easy to implement CapsNets due to the similar structure between them. Note that CapsNets consist of only Conv layers and capsule layers. Given that ResNet-20 [13] consists of 19 Conv layers followed by the last average pooling and FC layers, we can build a CapsNet on top of it by replacing the last two layers by a primary capsule (PrimaryCaps) and fully-connected capsule (FCCaps) layers, respectively. However, SmallNORB dataset is much less complex than CIFAR-10/SVHN, and its training set is relatively small (16200 samples), we use a 7-layer CNN, which is a shallower network with no shortcut connection. It consists of 6 CONV layers, followed by AvgPool and FC layers.

We also measure the performance variations according to the depth and width of capsule layers. The depth means how many routing layers are inserted after the PrimaryCaps layer. Thus, the depth of

Table 1: Comparison of parameter counts (M), FLOPs (M), and error rates (%) between routing algorithms of CapsNets and CNN models. We use ResNet-20 as the base network. DR, EM and SR stands for dynamic [34], EM [16] and self-routing, respectively. The number following (method-) is the number of stacked capsule layers on top of Conv layers. All CapsNets have 32 capsules in each layer. We test each model 5 times with different random seeds. Error rates reported below are their averages.

| Methods | # Param. (M) | # FLOPs (M) | CIFAR-10 | SVHN |
|---------|--------------|-------------|----------|------|
| AvgPool | 0.3 | 41.3 | $7.94_{\pm 0.21}$ | $3.55_{\pm 0.11}$ |
| Conv | 0.9 | 61.0 | $10.01_{\pm 0.99}$ | $3.98_{\pm 0.15}$ |
| DR-1 | 5.8 | 73.5 | $8.46_{\pm 0.27}$ | $3.49_{\pm 0.69}$ |
| DR-2 | 4.2 | 232.1 | $\mathbf{7.86}_{\pm 0.21}$ | $3.17_{\pm 0.09}$ |
| EM-1 | 0.9 | 76.6 | $10.25_{\pm 0.45}$ | $3.85_{\pm 0.13}$ |
| EM-2 | 0.8 | 173.8 | $12.52_{\pm 0.32}$ | $3.70_{\pm 0.35}$ |
| SR-1 | 0.9 | 62.2 | $8.17_{\pm 0.18}$ | $3.34_{\pm 0.08}$ |
| SR-2 | 3.2 | 140.3 | $\mathbf{7.86}_{\pm 0.12}$ | $\mathbf{3.12}_{\pm 0.13}$ |

1 means the final layers of the network are PrimaryCaps+FCCaps, between which the routing is performed once. The depth of 2 indicates the final layers are PrimaryCaps+ConvCaps+FCCaps so that the routing is performed twice between consecutive capsule layers. Therefore, the depth of $d$ involves $d - 1$ ConvCaps inserted between PrimaryCaps and FCCaps. All ConvCaps layers have a kernel size of 3 and a stride of 1 except for the first ConvCaps layer that has a stride of 2. The width indicates the number of capsules in each capsule layer; for example, the width of 8 means there are 8 capsules in all Primary/ConvCaps layers of the architecture.

**CNN baselines**. We also test two variants of CNNs for comparison between CapsNets and conventional CNNs. Since the former CONV layers of the base networks are shared by all the models, we vary the last two layers as (1) AvgPool+FC as the original architecture and (2) Conv+FC for verifying whether the performance obtained by CapsNets is simply caused by using more parameters, not by their structures or routing algorithms.

We describe more details of experiments in the Appendix, including architectures and optimization.

## 5.2 Results of Image Classification

We compare the image classification performance of self-routing with two agreement-based routing algorithms on SVHN [42] and CIFAR-10 [22]. Table 1 summarizes the error rates as well as memory and computation overheads of each method. Self-routing and dynamic routing [34] show similar classification accuracies to CNN baselines, while EM routing [16] degrades the performances. Importantly, the computation overheads of self-routing in FLOPs are less than those of other routing baselines, since it requires no iterative routing computations. In terms of the parameter size, EM routing is the most efficient due to its matrix representations. Yet, we find that it is hard to train the networks with stacked EM routing layers on datasets more complex than initially tested ones [16] (*e.g.* SmallNORB). It seems that the constraint of $4 \times 4$ weight matrix multiplications is too strong to learn good representations from complex data with multi-level routing layers. Dynamic routing is comparable to our self-routing in performance, but it requires much more FLOPs for computation. We find that CapsNet variants equipped with self-routing are better than CNN baselines, but the margins are not large. It seems that the benefit of using the ensemble of weak submodules is relatively weak since the degree of required specialization is small for the task and the datasets. In fact, MoE-style deep models are commonly strong in the tasks that require obvious specializations such as multi-task learning [2, 33, 30].

## 5.3 Robustness to Adversarial Examples

Adversarial examples are the inputs that are intentionally crafted to trick recognition models into misclassification. Numerous defensive methods for such attacks have been suggested [6, 28]. In [16], CapsNets have shown considerable robustness to such attacks without any reactive modification. Therefore, we evaluate our model's robustness to adversarial examples. We use the targeted and untargeted white-box Fast Gradient Sign Method (FGSM) [10] to generate adversarial examples. FGSM first computes the gradient of the loss with respect to the input pixels and adds the signs of the

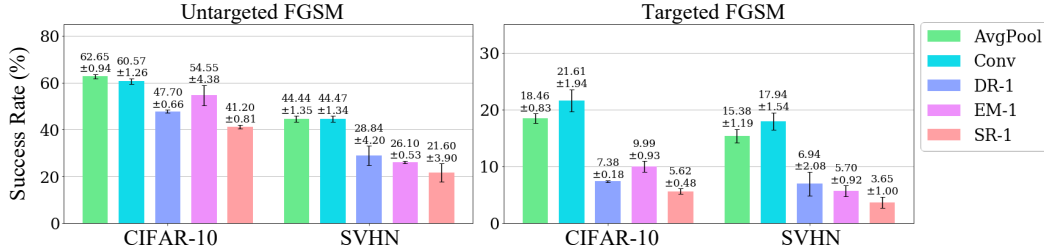

(a) PrimaryCaps+FCCaps (depth of 1)

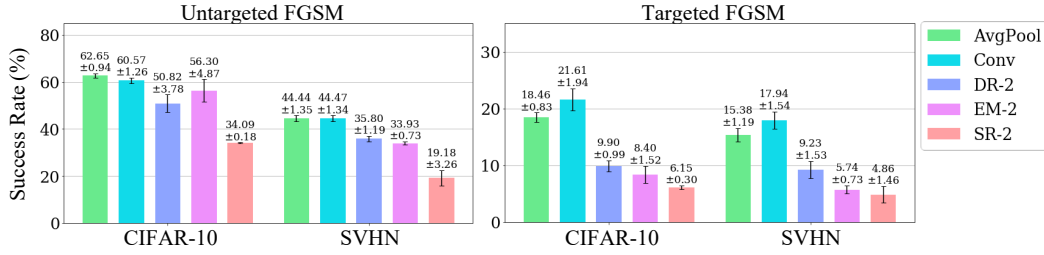

(b) PrimaryCaps+ConvCaps+FCCaps (depth of 2)

Figure 3: Success rates (%) of untargeted and targeted FGSM attacks against different routing methods of CapsNets and CNN models. All CapsNets have 32 capsules in each layer. We set $\epsilon = 0.1$ for all FGSM attacks. All results are obtained with 5 random seeds.

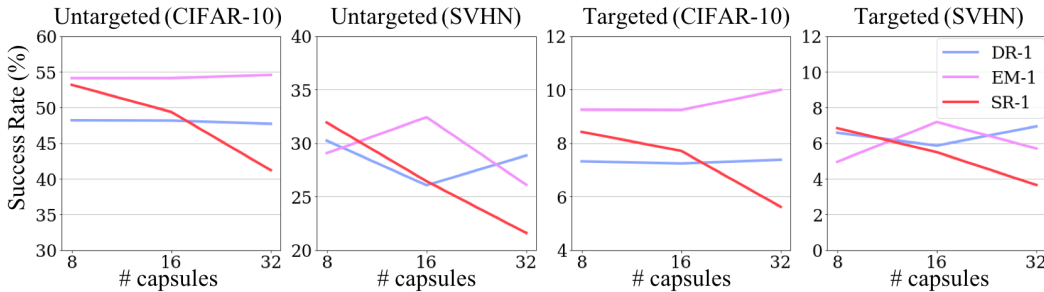

Figure 4: The variation of success rates of untargeted and targeted FGSM attacks according to the number of capsules per layer. The self-routing improves robustness as the number of capsules increases. We set $\epsilon = 0.1$ for all FGSM attacks. All results are obtained with 5 random seeds.

obtained tensor to the input pixels by a fixed amount $\epsilon$. For a fair comparison, we attack images for which predictions of each model are correct.

Figure 3 summarizes the results. CNN baselines are significantly more vulnerable against both general and targeted FGSM attacks than the CapsNets, among which our SR-CapsNets are the most robust. Figure 4 shows that adding more capsules per layer further improves the robustness of our SR-CapsNets, while stacking more capsule layers helps against the untargeted attack only. We cannot find a similar pattern in other routing algorithms. Often, their performance degrades when more capsules are used. This suggests that the routing-by-agreement struggles to cluster predictions whose large portions involve noisy information.

We also attach the results obtained by another adversarial attack of BIM [23] in the Appendix.

### 5.4 Robustness to Affine Transformation

One of CapsNets' known strengths is their generalization ability to novel viewpoints [16]. To demonstrate this, we measure the classification performance on the SmallNORB [24] test sets with novel viewpoints. Following the experimental protocol of [16], we train all models on 1/3 of training data with azimuths of 0, 20, 40, 300, 320, 340 degrees, and test them on 2/3 of test data with other

Table 2: Comparison of error rates (%) on the SmallNORB test set with the 7-layer CNN as the base architecture. Familiar and Novel denote the results on the test samples with seen and unseen viewpoints during training, respectively. All CapsNets have 32 capsules in each layer. All results are obtained with 10 random seeds.

| Methods | Azimuth | | Elevation | |
|---|---|---|---|---|
| | Familiar | Novel | Familiar | Novel |
| AvgPool | $8.49\pm0.45$ | $21.76\pm1.18$ | $5.68\pm0.72$ | $17.72\pm0.30$ |
| Conv | $8.39\pm0.56$ | $22.07\pm1.02$ | $7.51\pm1.09$ | $18.78\pm0.67$ |
| DR-1 | $6.86\pm0.50$ | $20.33\pm1.32$ | $5.78\pm0.48$ | $16.37\pm0.90$ |
| EM-1 | $7.36\pm0.89$ | $20.16\pm0.96$ | $5.97\pm0.98$ | $17.51\pm1.52$ |
| SR-1 | $7.62\pm0.95$ | $\mathbf{19.86}\pm1.03$ | $5.96\pm0.46$ | $\mathbf{15.91}\pm1.09$ |

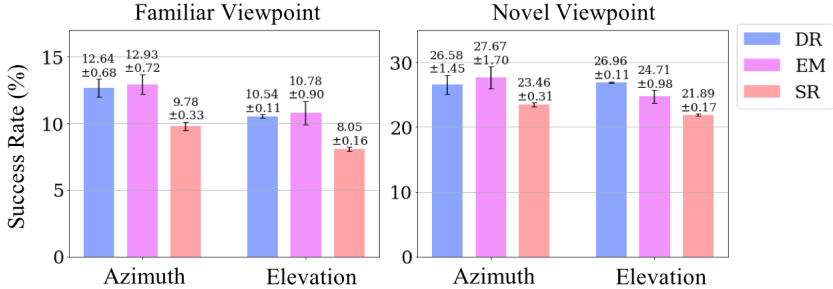

Figure 5: Comparison of error rates (%) on the SmallNORB test set without CNN base. All results are obtained with 5 random seeds.

azimuths. In another experiment, models are trained on 1/3 of training data with elevations of 30, 35, 40 degrees from the horizontal, and tested on 2/3 of test data with other elevations.

Table 2 shows the results where capsule-based models generalize better than the CNN baselines. Specifically, self-routing has the best performance for the images with novel azimuths and elevations. The results suggest that viewpoint generalization is not the unique strength of the routing-by-agreement. We also find a weak correlation between increase in model size (*i.e.* depth and width of capsule layers) and generalization performance, but the improvement is small.

Although the experiments with the 7-layer CNN base show that CapsNets generalize better than CNNs, the margins between different routing methods are not significant. Thus, we conduct additional experiments on SmallNORB with a smaller network that consists of only one convolution layer followed by three consecutive capsule layers of PrimaryCaps+ConvCaps+FCCaps. The capsule layers are composed of 16 capsules, each with 16 neurons. Figure 5 shows the results that our self-routing (SR) outperforms Dynamic and EM routing with significant margins in both tasks. That is, using shallow feature extractors, the previous routing techniques could struggle to learn good representations.

## 6 Conclusion

We proposed a supervised, non-iterative routing method for capsule-based models with better computational efficiency. We conducted systemic experiments for the comparison between the existing routing methods and our self-routing. The experiments verified that our method achieves competitive performance on adversarial defense and viewpoint generalization that are the two proposed strengths of CapsNets. Moreover, our method generally performs better when more capsules are used per layer, while the previous methods often behave unstably. The results suggested that the routing-by-agreement may not be a requirement for CapsNet's robustness. As future work, it is interesting to look for a method that can bring residual connections to our models, since it has been shown that residual networks behave like ensembles of networks with different depths [39]. It can be synergetic with our models where capsules take different paths of the same depth.

**Acknowledgements**. This work was supported by Samsung Research Funding Center of Samsung Electronics (No. SRFC-IT1502-51) and the ICT R&D program of MSIT/IITP (No. 2019-0-01309, Development of AI technology for guidance of a mobile robot to its goal with uncertain maps in indoor/outdoor environments). Gunhee Kim is the corresponding author.

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
