[Supplementary Material]

# [Appendix] Self-Routing Capsule Networks

**Taeyoung Hahn    Myeongjang Pyeon    Gunhee Kim**
Seoul National University, Seoul, Korea
{taeyounghahn,mjpyeon,gunhee}@snu.ac.kr

## A    Self-Routing Example

Figure 1: A simple example of the *self-routing* mechanism the number of capsules per layer is 2. See section 4.2 of the main paper for the details.

We visualize an example of self-routing in Figure 1. The routing coefficients are obtained by forwarding the input poses to routing networks, each of which has a similar role to a gating network in MoE. The routing coefficients are then multiplied by the input activations to obtain weighted votes. The output activations are the mean of these weighted votes. The pose predictions from input capsules are multiplied by the weighted votes and averaged to yield the output poses.

## B    Implementation Details

### B.1    Architectures

Table 1–2 describe the architecture of the 7-layer CNN used for SmallNORB [6] and the CapsNet headers for CIFAR-10 [4] and SVHN [9]. The header of Conv+FC is designed as a siamese of SR-1 with 32 capsules; it consists of a $3 \times 3$ Conv layer with $512(= 32 \times 16)$ channels, stride of 1, and $1 \times 1$ padding with a ReLU non-linearity and a FC layer with nclass activations. We use batch normalization [3] for fast training.

Table 1: The architecture of the 7-layer CNN used for SmallNORB.

| |
| --- |
| Conv $3 \times 3$, ReLU 16 stride 1, padding 1, BN |
| Conv $3 \times 3$, ReLU 32 stride 2, padding 1, BN |
| Conv $3 \times 3$, ReLU 32 stride 1, padding 1, BN |
| Conv $3 \times 3$, ReLU 64 stride 2, padding 1, BN |
| Conv $3 \times 3$, ReLU 64 stride 1, padding 1, BN |
| Conv $3 \times 3$, ReLU 128 stride 2, padding 1, BN |
| AvgPool $4 \times 4$ |
| FC 5 |

Table 2: The capsule layers that replace AvgPool and FC layers in the ResNet-20 and the 7-layer CNN. We below show the capsule layers at depth of 2. No ConvCap layer is used at depth of 1. `nc` and `nclass` denote the number of capsules per layer and the number of classes on the dataset, respectively. For PrimaryCaps and ConvCaps, we use $1 \times 1$ padding.

| Layer | Dynamic Routing | EM Routing | Self-Routing |
| --- | --- | --- | --- |
| PrimaryCaps | $3 \times 3$, stride 1<br>`nc` caps, dim 16<br>BN<br>Squash | $3 \times 3$, stride 1<br>`nc` caps, dim 16<br>BN<br>Sigmoid (act. only) | $3 \times 3$, stride 1<br>`nc` caps, dim 16<br>BN<br>Sigmoid (act. only) |
| ConvCaps | $3 \times 3$, stride 2<br>`nc` caps, dim 16<br>Squash | $3 \times 3$, stride 2<br>`nc` caps, dim 16<br>BN (pose only) | $3 \times 3$, stride 2<br>`nc` caps, dim 16<br>BN (pose only) |
| FCCaps | `nclass` caps<br>dim 16 | `nclass` caps<br>dim 16 | `nclass` caps<br>no pose |

## B.2 Optimization

We train all models using SGD optimizer for 350 epochs for CIFAR-10, 200 epochs for SVHN, and 100 epochs for SmallNORB. We set the initial learning rate to be one of 0.1, 0.01, and 0.001. We divide the learning rate by 10 at 150 and 250 epochs for CIFAR-10, at 100 and 150 epochs for SVHN, and at 50 and 75 epochs for SmallNORB.

For CIFAR-10 and SVHN, on which adversarial robustness is measured, we train all models with the cross-entropy loss, since the choice of a loss function is a nuisance factor in adversarial tests [8]. For SmallNORB, we use the margin loss [7] for dynamic routing, the spread loss [2] for EM routing, since they are the recommended losses in the original papers. We use the cross-entropy loss for all other models. In order to calculate the cross-entropy, we use the softmax function for CNN baselines, whereas we divide final activations by their sum instead of using the softmax for capsule baselines, since the activations are in $[0, 1]$. Our method guarantees the sum of activations in a location to be 1; hence no normalization is used. For faster training, we also use batch normalization [3] on augmented convolutional and Primary/ConvCaps layers with which the pooling layer is replaced. We do not use batch normalization on pose vectors of ConvCaps layers for dynamic routing and activation scalars of EM and self-routing, since their scales are enforced to be in $[0, 1]$. The number of iteration is set to 3 for both dynamic and EM routing. We use He uniform initialization [1] to initialize all weights except the routing networks, for which we set the initial weights to 0.

## C   More Results on White-Box Attacks

### C.1   FGSM

Table 3 reports additional results on FGSM attacks with more $\epsilon$ values.

### C.2   BIM

Table 4 reports results on Basic Iterative Method (BIM) [5], which applies FGSM multiple times with smaller step size. Again, all CapsNets outperforms the CNN baselines, and SR-CapsNets enjoy the best robustness among them. Note that the robustness of SR-CapsNet improves further as the number of capsules increases. It is consistent with the results obtained by FGSM in the main draft. We fix the number of iterations as 10 for all experiments.

Table 3: Success rates (%) of *untargeted* (top) and *targeted* (bottom) FGSM attacks against CapsNet and CNN models. The results are obtained with 5 random seeds.

| Methods | CIFAR-10 | | | | | | | | | SVHN | | | | | | | | |
|---|---|---|---|---|---|---|---|---|---|---|---|---|---|---|---|---|---|---|
| | $\epsilon = 0.1$ | | | $\epsilon = 0.2$ | | | $\epsilon = 0.3$ | | | $\epsilon = 0.1$ | | | $\epsilon = 0.2$ | | | $\epsilon = 0.3$ | | |
| AvgPool | 62.7 | | | 71.8 | | | 76.6 | | | 44.4 | | | 57.6 | | | 65.1 | | |
| Conv | 60.6 | | | 71.2 | | | 75.7 | | | 44.5 | | | 58.3 | | | 65.6 | | |
| | 8 | 16 | 32 | 8 | 16 | 32 | 8 | 16 | 32 | 8 | 16 | 32 | 8 | 16 | 32 | 8 | 16 | 32 |
| DR-1 | 48.2 | 48.2 | 47.7 | 57.1 | 58.6 | 57.5 | 64.1 | 66.5 | 65.1 | 30.2 | 26.1 | 28.8 | 39.1 | 34.2 | 37.3 | 46.6 | 42.8 | 45.2 |
| DR-2 | — | — | 50.8 | — | — | 59.8 | — | — | 67.1 | — | — | 35.8 | — | — | 49.0 | — | — | 57.9 |
| EM-1 | 54.1 | 54.1 | 54.6 | 65.3 | 66.4 | 65.9 | 72.1 | 72.5 | 71.4 | 29.1 | 32.4 | 26.1 | 43.5 | 47.1 | 39.9 | 54.6 | 57.3 | 52.3 |
| EM-2 | — | — | 56.3 | — | — | 69.2 | — | — | 77.6 | — | — | 33.9 | — | — | 48.8 | — | — | 58.5 |
| SR-1 | 53.2 | 49.4 | 41.2 | 64.0 | 59.2 | 51.7 | 70.7 | 66.4 | 60.8 | 31.9 | 26.4 | 21.6 | 44.7 | 36.1 | 31.2 | 54.1 | 44.9 | 41.9 |
| SR-2 | — | — | **34.1** | — | — | **45.9** | — | — | **56.9** | — | — | **19.2** | — | — | **28.1** | — | — | **39.0** |

| Methods | CIFAR-10 | | | | | | | | | SVHN | | | | | | | | |
|---|---|---|---|---|---|---|---|---|---|---|---|---|---|---|---|---|---|---|
| | $\epsilon = 0.1$ | | | $\epsilon = 0.2$ | | | $\epsilon = 0.3$ | | | $\epsilon = 0.1$ | | | $\epsilon = 0.2$ | | | $\epsilon = 0.3$ | | |
| AvgPool | 18.5 | | | 17.7 | | | 15.6 | | | 15.4 | | | 19.3 | | | 19.5 | | |
| Conv | 21.6 | | | 23.6 | | | 22.0 | | | 17.9 | | | 22.8 | | | 23.0 | | |
| | 8 | 16 | 32 | 8 | 16 | 32 | 8 | 16 | 32 | 8 | 16 | 32 | 8 | 16 | 32 | 8 | 16 | 32 |
| DR-1 | 7.3 | 7.2 | 7.4 | 7.7 | 7.6 | 7.8 | 7.8 | 7.9 | 8.1 | 6.6 | 5.9 | 6.9 | 8.4 | 7.3 | 8.8 | 9.3 | 8.0 | 9.6 |
| DR-2 | — | — | 9.9 | — | — | 9.6 | — | — | 9.2 | — | — | 9.2 | — | — | 11.9 | — | — | 12.7 |
| EM-1 | 9.3 | 9.2 | 10.0 | 9.2 | 9.6 | 10.2 | 9.5 | 9.4 | 9.3 | 5.0 | 7.2 | 5.7 | 6.5 | 10.2 | 7.1 | 8.0 | 11.0 | 8.4 |
| EM-2 | — | — | 8.4 | — | — | 8.7 | — | — | 8.5 | — | — | 5.7 | — | — | 6.9 | — | — | 8.2 |
| SR-1 | 8.4 | 7.7 | **5.6** | 8.6 | 7.8 | **6.1** | 8.6 | 8.1 | **6.7** | 6.8 | 5.5 | **3.7** | 9.2 | 7.1 | **4.9** | 10.3 | 8.0 | **6.2** |
| SR-2 | — | — | 6.2 | — | — | 7.1 | — | — | 7.4 | — | — | 4.9 | — | — | 6.9 | — | — | 8.3 |

Table 4: Success rates (%) of *untargeted* (top) and *targeted* (bottom) BIM attacks against CapsNet and CNN models. The results are obtained with 5 random seeds.

| Methods | CIFAR-10 | | | | | | | | | SVHN | | | | | | | | |
|---|---|---|---|---|---|---|---|---|---|---|---|---|---|---|---|---|---|---|
| | $\epsilon = 0.1$ | | | $\epsilon = 0.2$ | | | $\epsilon = 0.3$ | | | $\epsilon = 0.1$ | | | $\epsilon = 0.2$ | | | $\epsilon = 0.3$ | | |
| AvgPool | 84.9 | | | 93.2 | | | 95.5 | | | 62.2 | | | 79.8 | | | 86.2 | | |
| Conv | 82.0 | | | 93.1 | | | 95.8 | | | 62.6 | | | 80.1 | | | 86.9 | | |
| | 8 | 16 | 32 | 8 | 16 | 32 | 8 | 16 | 32 | 8 | 16 | 32 | 8 | 16 | 32 | 8 | 16 | 32 |
| DR-1 | 56.2 | 54.7 | 54.4 | 67.6 | 64.2 | 64.2 | 72.1 | 69.7 | 69.2 | 39.2 | 34.4 | 35.3 | 50.1 | 44.4 | 45.0 | 56.8 | 50.8 | 50.5 |
| DR-2 | — | — | 63.1 | — | — | 74.7 | — | — | 81.1 | — | — | 52.1 | — | — | 68.4 | — | — | 75.0 |
| EM-1 | 54.5 | 53.0 | 63.6 | 64.2 | 65.4 | 75.9 | 71.6 | 73.9 | 83.1 | 32.0 | 39.6 | 33.5 | 41.4 | 52.1 | 45.1 | 49.0 | 60.4 | 53.9 |
| EM-2 | — | — | 58.0 | — | — | 68.7 | — | — | 75.6 | — | — | 42.4 | — | — | 56.3 | — | — | 64.1 |
| SR-1 | 68.4 | 62.4 | 50.5 | 78.6 | 73.7 | **60.2** | 83.3 | 79.2 | **66.4** | 50.5 | 39.9 | **30.5** | 66.9 | 54.2 | **39.7** | 74.1 | 62.5 | **45.8** |
| SR-2 | — | — | **49.7** | — | — | 63.4 | — | — | 70.9 | — | — | 31.4 | — | — | 44.0 | — | — | 52.1 |
| AvgPool | 45.4 | | | 59.4 | | | 66.9 | | | 42.6 | | | 62.9 | | | 69.2 | | |
| Conv | 42.8 | | | 57.0 | | | 64.8 | | | 41.8 | | | 58.4 | | | 65.6 | | |
| | 8 | 16 | 32 | 8 | 16 | 32 | 8 | 16 | 32 | 8 | 16 | 32 | 8 | 16 | 32 | 8 | 16 | 32 |
| DR-1 | 17.7 | 15.9 | 17.3 | 27.4 | 24.5 | 26.6 | 34.0 | 31.0 | 33.1 | 18.5 | 17.6 | 16.6 | 30.3 | 28.3 | 27.5 | 38.5 | 36.0 | 35.0 |
| DR-2 | — | — | 21.2 | — | — | 26.7 | — | — | 28.8 | — | — | 23.4 | — | — | 36.2 | — | — | 43.5 |
| EM-1 | 19.0 | 18.0 | 24.1 | 27.6 | 27.4 | 35.9 | 33.8 | 33.7 | 44.0 | 11.0 | 19.2 | 16.0 | 41.4 | 52.1 | 45.1 | 23.7 | 39.2 | 35.5 |
| EM-2 | — | — | 18.5 | — | — | 28.2 | — | — | 30.7 | — | — | 14.5 | — | — | 35.2 | — | — | 32.7 |
| SR-1 | 28.1 | 21.8 | **12.1** | 40.7 | 33.3 | **17.1** | 48.5 | 40.1 | **20.6** | 23.3 | 15.2 | **8.6** | 39.8 | 26.1 | **13.8** | 50.1 | 34.1 | **17.9** |
| SR-2 | — | — | 15.0 | — | — | 23.9 | — | — | 29.8 | — | — | 15.3 | — | — | 27.8 | — | — | 36.0 |

## C.3 Generated Adversarial Images

We depict some examples of adversarial images generated by the untargeted and targeted FGSM attacks in Figure 2–3. No significant visual difference is observed.

(a) (b) (c) (d) (e) (f) (g) (h) (i) (j) (k) (l)

Figure 2: Generated adversarial images of (a) AvgPool+FC, (b) MaxPool+FC, (c) CONV+FC, (d) DR-1, (e) EM-1, and (f) SR-1 on CIFAR-10 and (g) AvgPool+FC, (h) MaxPool+FC, (i) CONV+FC, (j) DR-1, (k) EM-1, and (l) SR-1 on SVHN by the *untargeted* FGSM attack. We use ResNet-20 as a base network at $\epsilon = 0.1$. The results of CapsNets are obtained with the width of 32.

(a) (b) (c) (d) (e) (f) (g) (h) (i) (j) (k) (l)

Figure 3: Generated adversarial images of (a) AvgPool+FC, (b) MaxPool+FC, (c) CONV+FC, (d) DR-1, (e) EM-1, and (f) SR-1 on CIFAR-10 and (g) AvgPool+FC, (h) MaxPool+FC, (i) CONV+FC, (j) DR-1, (k) EM-1, and (l) SR-1 on SVHN by the *targeted* FGSM attack. We use ResNet-20 as a base network at $\epsilon = 0.1$. The results of CapsNets are obtained with the width of 32.