[Reviews · NeurIPS 2019]

Reviewer 1



This paper proposes a less principled but more successful alternative routing mechanism for capsule networks. It replaces the EM or dynamic algorithm with a simple perceptron model for predicting routing coefficients. This routing mechanism is thoroughly explored for its test performance, transformation robustness, as well as parameter and computational requirements. The results are convincing. The mechanism they propose improves test performance and robustness to adversarial attacks and view point transformations without significantly increasing parameters or computational cost. the paper is original. The routing procedure they propose is simple and somewhat self evident, but the improved performance they find as a result of the mechanism is impressive. And i know of no other work that has attempted to replace the routing mechanism with such a technique. The paper is well written though some elements could be made more clear (detailed in the improvements section) The paper is fairly significant as it explores a baseline for routing that i believe many researchers may have written off for being to simple, but demonstrates its success convincingly.

Reviewer 2



Post-rebuttal: I have considered the opinion and viewpoint of the other reviewers, who have both provided some good insight on the paper. I have also read the response of the authors very carefully, which has provided some more information. I am happy to revise my score reflecting the new evidence authors have provided. --------------------------------- ---Authors propose a new routing algorithm for CapsNets called Self-routing which is a supervised non iterative routing method that removes the agreement component from the routing process. Authors base their implementation on mixture of experts claiming a resemblance between a capsule and an expert. In that sense an expert is specialising in a different region of the input space, whose contributions are adjusted differently per example/input. What happens in the Dynamic Routing and EM is that the agreement between a higher level and lower level capsule is paramount for deciding if something is present in an image or which information to keep based on a voting process. This component is happening in an unsupervised manner. Authors claim that this property might be attributed to an ensemble (averaging) behaviour and not to the routing-by-agreement per se. I think this might require more experimental results than those presented in this paper. ---This paper presents some original results in terms of giving a different spin on the routing algorithms for CapsNets but it does not provide compelling arguments as to why this might be happening. Primarily I think the information pertaining to the how self-training process actual works in practise is rather limited, other than the fact that a pose vector is multiplied by a trainable weight matrix and outputs the routing coefficients. Maybe section 4.2 could include a step-by-step description of the process rather than relying on the mixture-of-experts process alone. ---The paper is clearly written but the different tables presenting various results take time to understand what they are referring to, e.g. parameters, error rates, etc. The experiments are comprehensive focusing on three different datasets and also the use of adversarial attacks. However some clarity on the point made in line 60 would be helpful. As far as I am aware neither dynamic routing nor EM routing papers mentions anything about using ResNet as part of their experiments with CapsNets other than defining a CNN baseline. Could authors elaborate a bit more on that? Is it the case they implemented Dynamic Routing and EM from scratch and then added ResNet in all implementations to enable comparisons with CIFAR-10 etc.? Is there a reason why the employed a CNN to do so instead of relying solely on CapsNets? Do they suggest that as it stands at the moment the best way forward is an ensemble of ResNet type models and CapsNets. ---The arguments and the quality of the work presented seem plausible to a large extent - but more elaboration on section 4.2 would help to improve the theory behind the concept of the new routing algorithm. ---CapsNets are a very promising area of research therefore new results, methods and insights are significant. The extend to which the methods presented in this paper are significant or novel is limited, but have an important spin towards changing the routing process not to include the agreement component. Could you please check the paper once more for correcting typos, e.g. robust instead of robustly in line 16, line 42 much more, etc.

Reviewer 3



Post rebuttal : The authors addressed the concerns I raised regarding error bars in the experiments. I am happy to update my rating to 7. -------------- This paper proposes a self-routing mechanism in which each capsule's pose goes through a little neural net which outputs the routing coefficients to each potential parent capsule. This is in contrast to previous work, where a key component of the model design was that routing of parts to wholes should be determined based on agreement of the pose of the whole across parts. The obvious drawback seems to be that if a part can reasonably belong to multiple wholes, it does not get the chance to choose one based on what other parts are agreeing on. Instead it must make a choice a-priori or just spread its vote across the many choices. However, it turns out that this is not detrimental, and in fact is advantageous, at least on tasks such as classification, adversarial robustness, and generalization to novel view points. Pros - The paper questions an important idea behind capsule networks. This helps answer the question whether it is routing by agreement that is important or just routing. - Experiments cover 3 different tasks. Cons - The results don't have error bars and based on all the numbers it seems that the improvement in the results may not be very significant. However, even matching the results is enough to make the point. Overall, the paper is interesting since it provides evidence that strong gating alone is an important factor that the community should pay attention to when designing variants of capsule networks. Alternatively, this suggests that the tasks that are being used for evaluating capsule networks are not making use of the agreement aspect of the routing. Either way the conclusions are interesting.

[Author Response · NeurIPS 2019]

¹ We thank the reviewers for their valuable feedback. The final version will resolve all the concerns raised by the reviews.

² [**Reviewer 1**]

³ **Experiments using stronger attacks**. During the rebuttal, we experiment the robustness to JSMA
⁴ attacks with 3 random seeds. As shown in the figure on the right, our approach is the most robust (*i.e.*
⁵ the lowest attack success rates) to JSMA attacks. We expect that CapsNets would be more robust to
⁶ gradient-based attack (FGSM, BIM, JSMA) than to optimization-based attack (DeepFool, CW), which
⁷ will be further discussed in the final draft.

⁸ **Depth and width of the capsule layers**. We fixed the routing iterations for dynamic and EM routing at 3, which yield
⁹ the best performance in their original papers. In this work, the depth means the number of capsule layers added to the
¹⁰ base network, and the width is the number of capsules per layer (the same across all capsule layers). We will clarify this.

¹¹ [**Reviewer 2**]

¹² **Experimental results for ensemble behavior**. In our initial experiments, we inspected the effect of agreement-routing
¹³ by training 3-layer dynamic and EM routing models (Conv→PrimaryCaps→FCCaps) on smallNORB dataset with
¹⁴ no viewpoint splits. At test, we then enforced fixed, uniform routing coefficients; as a result, each capsule does not
¹⁵ route to its best options anymore but uniformly spreads its influence to all upper-level capsules. Surprisingly, this
¹⁶ enforcement barely hurt the performance (DR: 91.48% to 90.76%, EM:86.52% to 85.70%). It partly hints that the effect
¹⁷ of agreement-routing could be questionable but the ensemble behavior might be sufficient.

¹⁸ **Step-by-step explanation on section 4.2**. We here summarize how self-routing works step-by-step: (1) Each capsule
¹⁹ multiplies its pose with $W_{pose}$ and $W_{route}$ (learnable parameters) to compute the predicted pose changes and the routing
²⁰ coefficients for the upper-level capsules, respectively. (2) Each capsule multiplies its activation output with the routing
²¹ coefficients to calculate the voting weights to the upper-level capsules. (3) We obtain the poses of upper-level capsules
²² by weighted-averaging the predicted pose changes of all lower-level capsules with their voting weights. (4) Finally,
²³ we compute the activations of upper-level capsules by weighted-averaging the routing coefficients of all lower-level
²⁴ capsules with their activations. In self-routing, if a capsule underperforms on some input data (*e.g.* ambiguous cases),
²⁵ the optimizer learns to assign less weight to the capsule as experts are trained in the MoE setting. This process is much
²⁶ easier in our method because they are directly supervised, whereas the effects of lower-level capsule activations to
²⁷ upper-level ones are indirect in previous agreement-based methods.

²⁸ **Why CNN baseline**. As previous CapsNets employed shallow and weak CNN base networks, we were curious whether
²⁹ the CapsNets can hold any advantage over recent deep networks (with residual connections). In our preliminary
³⁰ experiments, ResNet-20 outperformed the CapsNets even in viewpoint generalization tasks (smallNORB, affNIST)
³¹ where CapsNets were supposed to be stronger. Thus, we focused on verifying that employing capsule structures could
³² benefit the recent CNNs. It was our anecdotal motivation to employ ResNet as our base network. In the next question,
³³ we present the experimental results with no base network as R2 suggested.

³⁴ **Results with no CNN base**. During the rebuttal, we perform experiments on small-
³⁵ NORB viewpoint generalization tasks using a 4-layer architecture that consists of a
³⁶ convolution layer followed by 3 consecutive capsule layers (primary, convolutional and
³⁷ fully-connected). The primary and convolutional capsule layers consist of 16 capsules
³⁸ each and each capsule contains 16 neurons. The results are shown in the figures on the
³⁹ right. Each value denotes the mean of the error with 3 random seeds. Our self-routing
⁴⁰ (SR) outperforms Dynamic and EM routing with significant margins in both tasks. It shows that using shallow feature
⁴¹ extractors, the previous routing techniques struggle to learn good representations.

⁴² [**Reviewer 3**]

⁴³ **Error bars in the results**. We will add error bars to all the results in the final draft. As examples, we below show the
⁴⁴ mean and standard deviations of the performance with 10 random seeds for the experiments of adversarial attacks and
⁴⁵ novel viewpoints. Compared to other agreement-based routing CapsNets, our SR approach attains not only much lower
⁴⁶ errors but also lower or similar variances. All hyperparameters are set to be the same as in the original paper (*e.g.* we
⁴⁷ use 16 capsules per layer for smallNORB and 32 for CIFAR-10 and SVHN, and fix the depth as 1).

⁴⁸ **Tables are too dense**. We will replace some dense tables with graphs for better readability and visuality.

[Meta-Review · NeurIPS 2019]

The paper was reviewed by three experts who unanimously recommend to accept the paper. The author rebuttal was effective and improved the reviewers' scores. We look forward to seeing your work at the conference.